# Measurement Uncertainty in Clinical Validation Studies of Sensors

**DOI:** 10.3390/s23062900

**Published:** 2023-03-07

**Authors:** John Mark Ansermino, Guy Albert Dumont, Amy Sarah Ginsburg

**Affiliations:** 1British Columbia Children’s Hospital Research Institute, The University of British Columbia, Vancouver, BC V6T 1Z6, Canada; 2Department of Electrical and Computer Engineering, British Columbia Children’s Hospital Research Institute, The University of British Columbia, Vancouver, BC V6T 1Z8, Canada; 3Clinical Trials Center, University of Washington, Seattle, WA 98105, USA

**Keywords:** measurement, uncertainty, device validation, sensors, accuracy, precision, agreement, bias, reference device, clinical decision-making

## Abstract

Accurate clinical sensors and devices are essential to support optimal medical decision-making, and accuracy can be demonstrated through the conduct of clinical validation studies using validated reference sensors and/or devices for comparison. Typically unmeasurable, the true reference value can be substituted with an accepted physiological measurement with an associated uncertainty. We describe a basic model of measurement uncertainty that specifies the factors that may degrade the accuracy of an observed measurement value from a sensor, and we detail validation study design strategies that may be used to quantify and minimize these uncertainties. In addition, we describe a model that extends the observed measurement uncertainty to the resultant clinical decision and the factors that may impact the uncertainty of this decision. Clinical validation studies should be designed to estimate and minimize uncertainty that is unrelated to the sensor accuracy. The contribution of measurement observation uncertainty to clinical decision-making should be minimized but also acknowledged and incorporated into the clinical decision-making process.

## 1. Introduction

The massive global increase in the production and use of biomedical sensors and devices in both healthcare and at home due to the COVID-19 pandemic has raised concerns about the lack of accuracy of some of these sensors and devices. This concern was amplified by numerous reports of racial bias in pulse oximetry devices that continue to be investigated by governments and regulators [1,2,3]. Appreciating the inherent uncertainty in sensor measurement is critically important to improving both sensor and device accuracy and clinical decision-making. 

Clinicians and patients have the reasonable expectation that physiological measurements from a medical sensor or device are accurate. The interpretation of accuracy has changed over the last few years. Initially, accuracy was limited to an expression of bias; however, the current International Organization for Standardization (ISO) recommendation defines the accuracy of physiological measurements by both trueness (closeness of agreement between the arithmetic mean of a large number of test results and the true or accepted reference value) and precision (closeness of agreement between test results) [4]. To meet this expectation of accuracy, new sensors or devices can be compared to an existing reference sensor or device; however, it is first critical to understand the measurement uncertainty of the reference sensor or device. 

In many situations, the “true” underlying physiological value of a measurement may be ephemeral, time-varying, and unmeasurable, and the reference sensor or device itself will lack both measurement trueness and precision compared to the actual true value. Many applications of sensor measurements assume a deterministic model that applies average measurements derived from large populations to an individual, with no consideration of random variations. In reality, most sensor measurements are stochastic in which random variations, with some associated probability distribution, are inevitable [5]. The resulting uncertainty in measurement due to these random variations also needs to be recognized by users of the sensor or device when utilizing the measurements to make clinical decisions. Thus, during clinical validation studies when comparing to a reference sensor or device, effort to quantify measurement uncertainty should be undertaken.

## 2. Standards of Measurement

The science of measurement and its application, metrology, have attempted to standardize the basic principles and standards for performing device comparison studies. In 1997, the Joint Committee for Guides in Metrology (JCGM) [6], chaired by the Director of the Bureau International des Poids et Mesures [7], was formed by seven international standards organizations. The JCGM with the ISO have prepared the “Guide to the Expression of Uncertainty in Measurement” (GUM) [8] and the “International Vocabulary of Basic and General Terms in Metrology” (VIM) [9] along with technical working groups that provide regular updates to these documents. These standards are critically important for sensor and device validation studies. Without them, confusion can result from the use of the same terms but with different meanings or interpretations. Therefore, we would strongly recommend the use of the standardized vocabulary, definitions, and methodologies described in these GUM and VIM documents [8,9].

### 2.1. Uncertainty and Measurement Models

Every sensor or device, whether investigational or reference, will have some measurement error. The error may be a systematic error resulting in bias, or a random error. The use of the GUM “uncertainty approach” provides a method to characterize the quality of the measurement, accounting for both systematic and random errors, through the concept of measurement uncertainty [8]. Measurement uncertainty is dependent on multiple uncertainty factors in addition to sensor or device performance (Table 1). These uncertainty factors include study participant characteristics, the observer, the clinical context and setting, and the data capture and analysis methods. Uncertainty can be attributed to each of these factors based on the experimental design (e.g., repeat observations) and the expected distribution of the uncertainty (i.e., systematic vs random). Validation studies optimally should be designed to reduce the uncertainty due to these factors, while also modeling the contribution of each factor, especially in comparison to the uncertainty in the sensor or device measurement. A better and more nuanced understanding of the contribution of uncertainty factors to overall measurement uncertainty is essential to optimizing sensor performance while at the same time mitigating against poor validation metrics unrelated to sensor performance. The uncertainty approach also incorporates an acknowledgement that the true underlying physiological value is unknown but can be expressed in terms of a probability based on a degree of belief that this true value is being expressed [9]. In practice, a measurement model is used to describe the relationship between the uncertainty factors and the true value. Correction terms may also be included in the model when the uncertainty factors change over time or between experiments [9].

#### 2.1.1. Proposed Uncertainty Model for Sensor Measurement Validation Studies

The goal of modeling uncertainty is to better understand and account for the various uncertainty factors. A measurement validation study design can also be optimized to limit uncertainty. Specific contributions to uncertainty can be identified and quantified by changing one uncertainty factor and keeping the other factors consistent. Sample size will also impact uncertainty estimates. Simulation studies can be used to ensure that a sample is representative of the larger population, considering the other uncertainty factors. The proposed uncertainty model would be:Observed measurement value=true measurement value+sensor uncertainty+participant uncertainty+ observer uncertainty+context and setting uncertainty+data capture uncertainty+ analysis uncertainty+other uncertainty

Sensor uncertainty represents performance of a sensor or device. This is the key metric of interest in a clinical measurement validation study. Sensor characteristics such as stability, signal quality, and data averaging can significantly influence sensor uncertainty.Participant uncertainty relates to the normal and natural variability in an individual study participant’s physiological function. With physiological measurements, it is common for the true value to change during the measurement period, even in the same study participant. This uncertainty will depend on the sampling frequency, averaging interval, filtering, and measurement interval. Aliasing may also be present if the sample frequency is too low. Participant uncertainty can be qualified by changing the sampling frequency, averaging interval, filtering, and measurement interval for the same participant and observation.Observer uncertainty relates to the ability of an observer to achieve identical measurements under identical conditions on the same participant. Typically defined as repeatability, there is an interval between measurements, but it is assumed that conditions do not change within this short interval. Observer uncertainty can be quantified and reduced by training and experience, and when the same observer performs repeat observations within a short time interval.Context and setting uncertainty includes clinical factors, such as age, sex, ethnicity, disease severity, and sensor location, and environmental factors, such as temperature, light, sound, vibration, and movement (e.g., talking and wind), which can create additional uncertainty. Context uncertainty can be estimated by changing a single factor and evaluating the impact on the observations while minimizing all other uncertainty factors.Data capture uncertainty relates to the method used to capture measurements which can significantly increase uncertainty. Number preference is frequently observed when a user records a value from a sensor [10]. Synchronization of the investigational and reference sensors or devices is critically important to reduce uncertainty due to study participant uncertainty [11]. This is particularly critical in measurements with significant time variation. As perfect synchronization is not feasible, data capture uncertainty can be evaluated with the introduction of random fixed time delays. Data capture via automated digital recording is preferred for measurement validation studies to avoid number preference and ensure robust and consistent synchronization.Analysis uncertainty includes down sampling, averaging, and rounding. The use of fixed times of observations, such as breaths or beats per minute, are commonly not recognized as rounding down. Consistent methods of down sampling and averaging should be used to compare investigational and reference devices using as high a level of precision as possible to minimize analysis uncertainty. Using precise inter-breath or beat interval is preferable to counting the number of breaths or beats within an interval. Analysis uncertainty also can be evaluated using different methods on the same participant and observation.Other uncertainty relates to the reality that there may be other unexplained and unmeasurable sources of uncertainty, and all efforts should be made to identify and quantify these if possible.

In many clinical measurement validation studies, it is impractical, unfeasible, or impossible to estimate the contribution of each uncertainty factor; however, all potential uncertainty factors need to be considered in deciding if a comparison between sensor or devices is clinically acceptable. Choosing a measurement validation threshold that falls within study participant and/or observer uncertainties will compromise the sensor or device validation, which in turn, may lead to wasted effort, resources, and costs in performing the validation study, and even more concerning, may provide incorrect feedback to sensor/device developers regarding the sensor/device performance. Clinical validation studies should clearly identify and describe the uncertainty factors and the attempts made to quantify these factors based on the study design or previous studies performed under identical conditions. Regulatory, governmental, and nongovernmental organizations need to stipulate uncertainty factors in regulatory requirements and target product profiles to ensure robust comparisons between investigational and reference devices. The uncertainty must also be expressed using a valid and consistent measure of uncertainty. The reporting of an uncertainty of ± x% provides limited reproducibility as it unclear if this is a root mean square deviation, standard deviation, or limit of agreement. The statistic used to express uncertainty factors should be specified.

#### 2.1.2. Uncertainty in Clinical Decision-Making

A clinician’s decision, such as to treat or not treat or to refer or not refer, is typically based on a single observation at a single point in time using a generally agreed upon threshold value. Also typically, this threshold value is derived from a population-based measure of outcome. The uncertainty in the measurement and in the reference population outcome is rarely considered. A small change in a clinical observation, such as an oxygen saturation going from 89% to 90% can dictate a radically different treatment option while it represents a very small physiological change and is well within the uncertainty of both the measurement and the outcome. The uncertainty of the threshold value in addition to the sensor uncertainty cannot be ignored when making clinical decisions.
Observed measurement value = true measurement value + threshold uncertainty + sensor uncertainty+ participant uncertianty + observer uncertainty + context and setting uncertainty+ data capture uncertainty + anlaysis uncertainty + other uncertainty

Furthermore, the underlying clinical knowledge that is used to make clinical decisions also has uncertainty. This could be expressed as:Clincial decision=perfect knowledge+knowledge  uncertainty+true measurement value+ mesurement uncertainty

### 2.2. Improving Sensors and Devices to Reduce Uncertainty

Sensor and device designers can reduce uncertainty in clinical decision making by improving the sensor performance using data with a minimal amount of uncertainty and by designing devices that avoid or reduce uncertainty factors. For example, using forcing functions (such as repeat observations) to reduce within subject variability or rejecting measurements that are very likely to be artifacts. Real-time feedback can be provided to users to promote optimal measurement practices. Device interfaces should also clearly communicate the uncertainty associated with each measurement.

### 2.3. Improving Clinical Decision-Making

Educating healthcare providers to the relevance and importance of uncertainty in sensor and device measurement is key to improving the use and application of these technologies and ultimately, to clinical decision-making. Bias in human decision-making is a significant contributor to medical error that is compounded by measurement uncertainty, which is typically underestimated if considered at all. Education should be combined with training to reduce measurement uncertainty. Uncertainty should also be expressed in clinical guidelines by providing a clear indication of the data source for the guideline threshold and providing an expression of uncertainty (such as a confidence interval) for the threshold. The clinical decision-making process could be enhanced by delivery of data and information though features such as visualizations with confidence bounds. Increasing use of measurements and clinical data to drive individualized, precision health will likely increase the frequency of medical errors if we do not make efforts to recognize and reduce measurement uncertainty factors. This will be especially true as we use more sophisticated methods such as machine learning to analyze and integrate sensor and device data. We propose that sensor and device clinical validation studies be designed to minimize uncertainty and to address uncertainty factors in the context of measurement observations. Every effort should be made to measure uncertainty factors and to weigh these against the thresholds used for clinical decisions. Improved clinical decision-making and outcomes can be accomplished by reducing uncertainty, and with a better appreciation of the uncertainty, improved training and better sensor and device design can also be achieved.

## Figures and Tables

**Table 1 sensors-23-02900-t001:** Measurement uncertainty factors.

Uncertainty Factor	Example	Study Design to Measure Uncertainty	Optimization
Measurement uncertainty			
Sensor	Sensor performance variability	Repeatability with multiple sensors on a single subject at the same time	Improve sensor quality control
Subject	Within subject variability	Sample over an extended period of time; avoid aliasing	Ensure subject is in a stable state; reduce external perturbations
Observer	Within observer variability	Repeatability with multiple measurements on multiple participants	Enhanced observer training, larger measurement sample, repeat observations or device forcing functions
Context	Site, age, or disease state variability	Repeatability at different physical sites, in subjects of different ages, for different conditions or diseases	Adjust sensor and algorithms based on context
Data capture	Number preference or low precision	Compare manual vs. electronic data collection; compare precision thresholds	Electronic data collection and maximum precision (two decimal places)
Analysis	Averaging of results or counting events	Compare averaging methods (e.g., mean, median, mode, filter); compare counting with event intervals	Standardize averaging; use event interval
Unknown	Uncertainty present that cannot be removed by optimizing other causes of uncertainty	Optimize all other sources of uncertainty	Not possible to optimize without identifying source
Clinical decision uncertainty			
Threshold	Threshold based on expert opinion	Compare threshold to robust patient outcome	Threshold based on robust patient outcome
Knowledge	Decision based on expert opinion	Compare decision to robust patient outcome	Decision based on robust patient outcome

## Data Availability

Not applicable.

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
