# Peer review of "Measurement Uncertainty in Clinical Validation Studies of Sensors"

_sensors, 2023, doi:10.3390/s23062900_

Round 1

Reviewer 1 Report

This is a nicely written article that can be used as guidance for interested researchers. 

It can be suggested that methodological references can be added (statistical approaches and relevant software). 

References can include Bland-Altman analysis and the linear model extensions proposed by Taffe (e.g., https://journals.sagepub.com/doi/abs/10.1177/0962280216666667 and similar). 

Implementation using Stata, SAS, MedCalc or R (via relevant packages) 

Author Response

Ms. Teodora Đurić
Assistant Editor, MDPI Beograd
E-Mail: teodora.djuric@mdpi.com

Dear Ms. Đurić,

 We sincerely thank the reviewers for taking the time to review our submitted manuscript. Please note that we intended this invited manuscript to be reviewed as a “Perspectives” and not as a research article. It is our understanding that Perspectives submissions are “personal points of view on the state-of-the-art of a specific area of knowledge and its future prospects.”  In our letter to the editor, we elucidated our concerns about the lack of robust methodology for considering uncertainty in many validation studies. With this understanding, please see below a point-by-point response to the reviewers’ comments.

Best regards,

Mark Ansermino, on behalf of the co-authors

Reviewer 1

This is a nicely written article that can be used as guidance for interested researchers. 

It can be suggested that methodological references can be added (statistical approaches and relevant software). 

References can include Bland-Altman analysis and the linear model extensions proposed by Taffe (e.g., https://journals.sagepub.com/doi/abs/10.1177/0962280216666667 and similar). 

Implementation using Stata, SAS, MedCalc or R (via relevant packages) 

Authors’ response: Unfortunately, no single statistical approach is appropriate for every different specific validation study. The chosen statistical approach is highly dependent on the context, reference standard, and purpose. We are not aware of any software tools that would enable a researcher to model uncertainty reliably.

The work by Taffe, Nawarathna, Choudhary, and others on heteroscedastic regression lines with Bland Altman plots with replicate observations is critically important where these plots are used for validation reporting. However, we think this is outside the scope of this manuscript which is describing a more general approach to measurement uncertainty.

Reviewer 2 Report

This is a pioneer study resulting uncertainty in measurement due to these random variations also needs to be recognized by users of the sensor or device when utilizing the measurements to make clinical decisions. They found that sensor and device clinical validation studies are designed to minimize uncertainty and to address uncertainty factors in the context of measurement observations.

This is an interesting study with some new findings in this area of research. However, I nevertheless have the following comments that required to be addressed.

1.     The study design should be specified in title of this study. The authors should clarify this concern.

2.     Does any statistical methods used in this study?

3.     Does any simulation conducted in this study?

4.     For tables, I suggested to add some tables of summary of previous relevant studies.

5.     Lastly, some references should be updated.

Author Response

Ms. Teodora Đurić
Assistant Editor, MDPI Beograd
E-Mail: teodora.djuric@mdpi.com

Dear Ms. Đurić,

We sincerely thank the reviewers for taking the time to review our submitted manuscript. Please note that we intended this invited manuscript to be reviewed as a “Perspectives” and not as a research article. It is our understanding that Perspectives submissions are “personal points of view on the state-of-the-art of a specific area of knowledge and its future prospects.”  In our letter to the editor, we elucidated our concerns about the lack of robust methodology for considering uncertainty in many validation studies. With this understanding, please see below a point-by-point response to the reviewers’ comments.

Best regards,

Mark Ansermino, on behalf of the co-authors

Reviewer 2:

This is an interesting study with some new findings in this area of research. However, I nevertheless have the following comments that required to be addressed.

  1. The study design should be specified in title of this study. The authors should clarify this concern.

Authors’ response: We would like to clarify that in this Perspective, which is meant to relay our personal points of view on the state-of-the-art of measurement uncertainty in clinical validation studies of sensors and future prospects, we are not describing a specific study, but rather, an approach to all clinical validation studies.

  1. Does any statistical methods used in this study?

Authors’ response: We do not describe any specific statistical methods in this Perspective; we describe an approach to an error model that could be used in a validation study.

  1. Does any simulation conducted in this study?

Authors’ response: No, in this Perspective, we are not describing a specific study, but rather a general approach.

  1. For tables, I suggested to add some tables of summary of previous relevant studies.

Authors’ response: The purpose of this Perspective is not to describe (or critique) previously published studies, but rather to highlight the importance of the different sources of measurement uncertainty in validation studies. We are at the page limit for this “Perspectives” type of manuscript.

  1. Lastly, some references should be updated

Authors’ response:  As suggested, we have checked and updated all references.

Round 2

Reviewer 2 Report

No further comments.